# Generation of Functional Immortalized Human Corneal Stromal Stem Cells

**DOI:** 10.3390/ijms232113399

**Published:** 2022-11-02

**Authors:** Aurelie Dos Santos, Ning Lyu, Alis Balayan, Rob Knight, Katherine Sun Zhuo, Yuzhao Sun, Jianjiang Xu, Martha L. Funderburgh, James L. Funderburgh, Sophie X. Deng

**Affiliations:** 1Stein Eye Institute, University of California Los Angeles, Los Angeles, CA 90095, USA; 2Department of Ophthalmology and Visual Science, Eye & ENT Hospital, Shanghai Medical College of Fudan University, Shanghai 200031, China; 3Human Biology Society, University of California Los Angeles, Los Angeles, CA 90095, USA; 4Department of Ophthalmology, The First Affiliated Hospital of China Medical University, Shenyang 110001, China; 5Eye and Ear Institute, University of Pittsburgh, Pittsburgh, PA 15260, USA; 6Molecular Biology Institute, University of California Los Angeles, Los Angeles, CA 90095, USA

**Keywords:** corneal stromal stem cells, immortalization, bioengineered cell line

## Abstract

In addition to their therapeutic potential in regenerative medicine, human corneal stromal stem cells (CSSCs) could serve as a powerful tool for drug discovery and development. Variations from different donors, their isolation method, and their limited life span in culture hinder the utility of primary human CSSCs. To address these limitations, this study aims to establish and characterize immortalized CSSC lines (imCSSC) generated from primary human CSSCs. Primary CSSCs (pCSSC), isolated from human adult corneoscleral tissue, were transduced with ectopic expression of hTERT, c-MYC, or the large T antigen of the Simian virus 40 (SV40T) to generate imCSSC. Cellular morphology, proliferation capacity, and expression of CSSCs specific surface markers were investigated in all cell lines, including *TNFAIP6* gene expression levels in vitro, a known biomarker of in vivo anti-inflammatory efficacy. SV40T-overexpressing imCSSC successfully extended the lifespan of pCSSC while retaining a similar morphology, proliferative capacity, multilineage differentiation potential, and anti-inflammatory properties. The current study serves as a proof-of-concept that immortalization of CSSCs could enable a large-scale source of CSSC for use in regenerative medicine.

## 1. Introduction

Mesenchymal stem cells (MSCs) derived from the limbal stroma, termed corneal stromal stem cells (CSSCs), can be readily isolated from human corneal button donations and limbal biopsies [1]. CSSCs offer a remarkable model for the development of therapies due to their MSCs properties including immunomodulatory activity and multilineage differentiation potential including differentiation into corneal keratocytes [2,3]. Additionally, CSSCs demonstrated a regenerative capacity for the treatment of corneal epithelial injuries, inflammation, and corneal stromal opacities [1,4,5,6]. Treatment of corneal defects or injury models using CSSCs restores corneal transparency in animal wounding models [1,7].

The use of animals is still the predominant requirement in ocular surface research, especially in mechanistic and drug testing studies. These in vivo models are labor intensive and time consuming and may not reflect the function of the test article in humans [8,9,10]. Cell-based models present potential in advancing drug and cell-based therapeutic discovery for corneal opacities more rapidly, but current ocular surface research models are predominantly animal models. Therefore, the development of drug and cell-based therapies for corneal opacities is hampered by the lack of in vitro cell-based models.

While cellular models could serve as quick and efficient tools for ocular surface research, they present a major challenge starting with finding the adequate model. MSCs, including CSSCs, have greater potential as models over other types of cells, but their application in drug development is hindered by their short lifespan in culture accompanied by a loss of their multipotency and regenerative properties caused by the shortening of telomere length and replicative senescence from the first passage onward [11,12,13].

While the idea of immortalizing primary cells is not new, there have been several reports outlining the difficulty in producing a cell line that maintains the phenotype of the primary MSCs [14,15,16,17]. What is novel here is the successful immortalization of CSSCs that maintain the function of the primary cells while also demonstrating the desired ability for continued cell culture without senescence beyond 10 passages.

In this study, to overcome the limitation of primary MSCs, we generated immortalized CSSCs lines (imCSSC) and compared their phenotypes to the primary CSSC (pCSSC) from which imCSSC were derived from. Three approaches of immortalization were compared: overexpression of the human telomerase reverse transcriptase (hTERT), the oncogene c-MYC, and the large T antigen of the Simian virus (SV40T).

## 2. Results

### 2.1. Generation of Immortalized CSSC

Newly generated immortalized CSSC (imCSSC) lines, namely hTERT-imCSSC, c-MYC-imCSSC, and SV40T-imCSSC, were characterized and compared to pCSSC. The gene expression levels of hTERT, c-MYC, and SV40T in each imCSSC line were evaluated by RT-PCR. All genes were highly expressed in their respective imCSSC (Figure 1B). High levels of hTERT transcript were detected in hTERT-imCSSC compared to that of c-MYC-imCSSC (7-fold difference, *p*-value < 0.0001), while not detected in pCSSC and SV40T-imCSSC (all *p*-value < 0.0001, Figure 1B). Similarly, c-MYC transcript expression was significantly higher in c-MYC-imCSSC compared to that of pCSSC (782-fold), hTERT-imCSSC (608-fold), and SV40T-imCSSC (228-fold; all *p*-values < 0.0001, Figure 1B). As expected, SV40T mRNA was only detected in SV40T-imCSSC (*p*-value < 0.0001, Figure 1B).

Parental pCSSC had a morphology typical of MSCs: a small, spindle-shaped cell body with short processes (passage 3, Figure 1C). A similar morphology was observed in c-MYC-imCSSC, SV40T-imCSSC, and hTERT-imCSSC cells at early passages following immortalization (passage 7, Figure 1C). At later passages, pCSSC transitioned to a fibroblast-like phenotype, with larger cell bodies and elongated cellular processes (passage 7, Figure 1C), whereas c-MYC-imCSSC and SV40T-imCSSC both maintained their morphology for over 15 passages (both p15; Figure 1C). Furthermore, c-MYC-imCSSC and SV40T-imCSSC overexpression was successful in bypassing replicative cell cycle arrest of pCSSC, as most of the cells (96% and 97%, respectively) were actively proliferating at passage 10 compared to 1% in pCSSC at passage 7 as detected by Ki67 expression (Figure 2). Although hTERT-imCSSC propagated for an additional six passages, there was a shift in their phenotype to a large and flat morphology (Figure 1C) accompanied by a reduced proliferation, which indicated cellular senescence. The overexpression of hTERT failed to overcome cellular senescence beyond passage 12 and this immortalization approach was deemed unsuccessful. The rest of the experiments were conducted comparing c-MYC-imCSSC and SV40T-imCSSC with respect to pCSSC.

### 2.2. Phenotypic Characterization of Immortalized CSSC

Furthermore, c-MYC-imCSSC and SV40T-imCSSC were investigated for their expression of MSCs surface antigens (CD90, CD73, and CD105) and lack of expression of negative markers (CD14, CD20, CD34, and CD40) [2,18]. The proportion cells expressing the positive markers (CD90, CD73, and CD105) were similar in the imCSSC and pCSSC (Figure 3A). The cumulative expression levels of all the negative markers were less than 5% in all imCSSC (Figure 3A).

To further characterize the imCSSC, we investigated their multilineage differentiation capacity in vitro towards adipogenic, osteogenic, and chondrogenic fates [18]. Following adipogenic lineage induction, lipid droplets and adiponectin protein expression were detected in all imCSSC and pCSSC (red, Figure 3B). This observation was further confirmed by an increased mRNA expression of adipogenic markers: fatty acid binding protein 4 (FABP4) and perilipin (PLIN) after 7 and 21 days of induction (Figure 3C, top panel) [19]. Higher basal expression levels of FABP4 and PLIN were observed in undifferentiated SV40T-imCSSC and c-MYC-imCSSC as compared to that of pCSSC (*p*-value < 0.05, Figure 3C, top panel).

Following osteogenic induction, both c-MYC- and SV40T-imCSSC demonstrated increased calcium deposits observed by Alizarin red staining compared to pCSSC (Figure 3B). To further confirm osteogenic lineage commitment, transcript levels of alkaline phosphatase (ALP) [20] and osteonectin (SPARC) [21] were assessed at days 7 and 21 of induction. ALP levels increased at day 7 and day 21 in pCSSC and SV40T-imCSSC (all *p* values < 0.05, Figure 3C, middle panel), while decreased in c-MYC-imCSSC (*p*-value = 0.0213 at day 7 and *p* value = 0.0008 at day 21).

Alcian blue staining of chondrocyte-associated extracellular proteoglycans confirmed chondrogenic lineage differentiation in all cell lines after 21 days of induction (Figure 3B). The induction was confirmed by analysis of aggrecan (ACAN) and cartilage oligomeric matrix protein (COMP) transcripts [22]. Both ACAN and COMP were upregulated after 7 days of induction in all cell lines (D7, all *p*-values < 0.05, Figure 3C, bottom panel).

To summarize, SV40T- and c-MYC-imCSSC preserved the typical pCSSC cell morphology. Both c-MYC- and SV40T-imCSSC maintained MSCs surface antigen expression profile and multilineage differentiation capacity. 

### 2.3. Keratocyte Lineage Commitment

We next investigated the impact of immortalization on the differentiation ability into keratocytes. After 7 days of keratocyte lineage induction with TGF-ß3 and FGF2 [2,4,23], the morphology and organization of the investigated cells demonstrated distinctive morphologic features (Figure 4A). Moreover, pCSSC were organized in a monolayer with long extended cellular processes, a characteristic of quiescent keratocytes (Figure 4A) [24]. Following the 7-day induction, SV40T-imCSSC were arranged in multiple layers of small cells. In contrast, c-MYC-imCSSC did not sustain confluency and maintained a small, round cell body similar to their undifferentiated cells (Figure 4A). The transcript levels of the keratocyte markers lumican (LUM), keratocan (KERA) [25,26], and prostaglandin D2 synthase (PTGDS) [27] were all upregulated in pCSSC and SV40T-imCSSC in each cell line after 7 days of induction, as compared to their undifferentiated state (D0) (*p* < 0.05, Figure 4C–E). Contrastingly, c-MYC-imCSSC upregulated the corneal progenitor marker PAX6 at day 7 of differentiation (*p*-value = 0.0002, Figure 4B), but not LUM, KERA, nor PTGDS [28]. Taken together, keratocyte lineage commitment was observed to a greater extent in SV40T-imCSSC.

### 2.4. Modulation of Inflammation

The anti-inflammatory effect of CSSCs, as measured by the secretion of TSG-6 upon stimulation in culture, correlates with their anti-inflammatory potential in an in vivo mouse model of corneal wounding [4,28,29]. The transduction, integration, and overexpression of c-MYC and SV40T genes may affect the functional properties of CSSCs. Therefore, we investigated the expression of *TNFAIP6* in pCSSC, SV40T-, and c-MYC-imCSSC after stimulation with pro-inflammatory cytokines IFN-γ and TNF-α. *TNFAIP6* was detected in all cell lines investigated. The basal levels in unstimulated c-MYC- and SV40T-imCSSC were similar to those in pCSSC (Figure 5). After stimulation with IFN-γ and TNF-α, *TNFAIP6* levels were significantly upregulated in pCSSC and SV40T-imCSSC with a 4.2- and 1.4-fold increase, respectively, as compared to their basal levels (all *p* values < 0.05). Although not significant, pro-inflammatory cytokines-treated c-MYC-imCSSC upregulated *TNFAIP6* with 1.8-fold (*p*-value = 0.101, Figure 5). Our results suggested that SV40T retained anti-inflammatory properties, although to a lesser extent than pCSSC did.

## 3. Discussion

With the goal of establishing a stable cell line for applications in research and drug development, we investigated the feasibility of immortalizing pCSSCs. Three approaches of immortalizing pCSSCs were investigated: overexpression of hTERT, overexpression of the oncogene c-MYC, and ectopic expression of SV40 large T antigen. This is the first report that demonstrates the immortalization of pCSSCs while also demonstrating the significant variations the different immortalization strategies have on the same starting cell population.

All newly generated cell lines, hTERT-imCSSC, c-MYC-imCSSC, and SV40T-imCSSC, extended the pCSSC lifespan. Moreover, hTERT overexpression bypasses replicative cellular senescence and prolongs cellular lifespan in vitro while retaining their progenitor/stem cells properties [17,30,31,32,33]. Overexpression of hTERT was not sufficient in extending CSSCs’ expansion beyond 12 passages [34,35]. Similar to our findings, others observed that hTERT alone was not sufficient to immortalize adipose and bone marrow derived-MSCs [36,37] but has been demonstrated to extend the lifespan of other cells such as oral progenitor cells [17]. The addition of other viral genes such as p53 [38], BMI-1 [39], or HPV E6/E7 [40] to hTERT overexpression has demonstrated to improve the efficacy of hTERT overexpression to produce immortalized cells. However, the combination of hTERT with other oncogenes increased chromosomal instability, although differentiation capacity in late passages up to 40 passages has been reported [41]. Hence, the incorporation of additional oncogenes to hTERT makes this approach suboptimal towards the goal of establishing a stable CSSC cell line.

Overexpression of either c-MYC or SV40T was successful in prolonging CSSC’s lifespan. Furthermore, SV40T- and c-MYC-imCSSC had comparable expression of MSCs markers and properties to those of pCSSCs defined by the International Society for Cell Therapy [18].

Ectopic expression of SV40T successfully immortalizes human MSCs derived from different tissues including bone marrow [42,43,44], umbilical cord [12], coronal sutures [45], cranial periosteum [46], dental follicle [11,47], and articular cartilage [48,49] while maintaining their features. The expression of SV40T promotes cell cycle progression in immortalized cells through the inhibition of p53 and retinoblastoma gene (Rb) [50,51]. Particularly, the large T antigen interaction with Rb proteins hampers the cell’s ability to regulate its proliferation from external signals such as cell confluence [51]. As a consequence, SV40T overexpression leads to indefinite cell cycle progression and overgrown cell culture as observed in SV40T-imCSSCs [51]. In addition to the capacity to express keratocyte lineage markers under appropriate differentiation induction [28], SV40T-imCSSC sustained their response to pro-inflammatory cytokines assessed by the expression levels of TSG-6 transcript, which is an indicator of anti-inflammatory and stromal regeneration of CSSC in vivo [29].

Overexpression of c-MYC promotes high proliferation rates in human MSCs derived from adipose tissue and bone marrow but attenuated their differentiation capacity [52]. Similarly, Kami and colleagues reported that bone marrow-derived MSCs transduced with c-MYC did not differentiate into adipocytes [53]. In our study, a decreased differentiation capacity of c-MYC-imCSSC compared to pCSSC was associated with a loss of cell adherence in culture under induction (2- and 3-dimension) as also reported by Chen et al. [54]. Overall, our findings are consistent with the literature. Interestingly, although not significant, a slight trend of reduction in the population of CD90^+^ cells was observed and a lower expression level of *ALP* upon osteogenic induction in c-MYC-imCSSC. CD90, or Thy-1, a glycosylphosphatidylinositol-anchored glycoprotein first described in mice T cells [55,56], might be associated with the undifferentiated state of MSCs [57]. CD90 was presumed as an effective differentiation marker in the development of osteoblasts, and as the osteoblast matures, its expression declines [58,59,60]. Wang and colleagues also reported a decrease of CD90 in c-MYC-immortalized human bone marrow-derived MSCs [61]. This observation was accompanied by the loss of self-renewal and differentiation capacity [61,62,63].

The generation of these immortalized CSSCs will allow for significant advancement in the generation and translation of ophthalmic drug research. The capability to investigate drug-cell interaction with a reliable, reproducible cell line that functions equivalently to the primary cells from which it is derived is vitally important. Not only will these immortalized cell lines be beneficial in the field of drug discovery, but they will also allow for further research in the regenerative and anti-scarring potential of CSSCs themselves. Without the need to continuously isolate primary cells, the regenerative potential of these immortalized CSSCs can also be investigated.

In conclusion, the present study demonstrates CSSC can be immortalized while retaining the characteristics and function of their parental CSSC after ectopic overexpression of c-MYC or SV40T. Selection of a clonal and stable imCSSC line could address the senescence limitation of primary CSSCs. Furthermore, the phenotype and functions of imCSSC in late passages need to be further investigated.

## 4. Materials and Methods

### 4.1. Cell Culture

Experimentation on human tissue adhered to the tenets of the Declaration of Helsinki. The experimental protocol was evaluated and exempted by the University of California, Los Angeles Institutional Review Boards. Consent was obtained by the eye banks for the tissues to be used for research. Human corneoscleral tissue from healthy donors aged 20 to 75 years old was used and no differences were found based on the gender or age of the tissue donors. Primary CSSCs were isolated from human adult corneoscleral rims and propagated as previously described [1,3,4]. Briefly, the dissected tissue, the superficial limbal epithelium, and about 2/3 of the corneal stroma was incubated in collagenase blend L type (Sigma Aldrich, Saint Louis, MO, USA) overnight at 37 °C. Primary CSSCs were cultured on FNC Coating Mix^®^-coated cell culture plasticware (FNC coating solution, AthenaES, Baltimore, MD, USA), with MCDB201/DMEM-low glucose-based medium supplemented with 2% human serum (Innovative Research, Inc., Perary Court Novi, MI, USA), 10 ng/mL epidermal growth factor (Sigma Aldrich), 10 ng/mL platelet-derived growth factor (PDGF-BB, R & D Systems, Minneapolis, MN, USA), insulin-transferrin-selenium (ITS) solution (0.1×; Life Technologies, Carlsbad, CA, USA), 0.1 mM ascorbic acid-2-phosphate (Sigma Aldrich), 100 mM dexamethasone (Sigma Aldrich), penicillin-streptomycin solution (1X; Life Technologies), and 50 μg/mL gentamicin (Life Technologies). Cells were passaged when they reached 80–90% confluency.

### 4.2. Generation of Immortalized CSSC Lines

Lentiviral vectors were generated at the UCLA Integrated Molecular Technologies Core (Figure 1A). Regarding c-MYC and the large T antigen of the Simian virus 40, plasmids for the lentiviruses were obtained from Addgene (pcDNA3-c-MYC, #16011 and pBSSVD2005, #21826, respectively, Watertown, MA, USA). Each gene was packaged in a lentivirus backbone with a puromycin resistance gene, pac. Primary CSSCs, from the same tissue donor, at passage 2 (50–60% confluence, 6-well plate well) were transduced by the appropriate viral particles with a final particle amount of 0.54–0.58 µg for 24 h. Cells were washed with PBS and incubated with fresh medium. Three days later, puromycin selection (1 µg/ml) was started for 15 days, with medium renewal every 3 to 4 days.

### 4.3. Multilineage Differentiation

Multilineage induction toward adipogenic, osteogenic, chondrogenic, and keratocyte fates were performed by using the MesenCult™ stimulatory and differentiation kits (Human, STEMCELL Technologies Inc., Vancouver, Canada). Primary and immortalized CSSCs were plated at a cell density of 10^4^/cm^2^ for adipogenic and osteogenic induction. When cells reached 70–80% confluence in culture, osteogenic differentiation was started by using the MesenCult™ Osteogenic Stimulatory Kit (Human, STEMCELL Technologies Inc.). Once multilayering of the cells was observed, β-glycerophosphate (3.5 mM) was added to the stimulatory medium; the day of this occurrence was defined as day 0 of differentiation. CSSC-derived osteoblasts were fixed at day 21 of differentiation using 2.5% glutaraldehyde solution for 20 min at room temperature (RT), washing twice with phosphate-buffered saline (PBS), and staining with Alizarin red solution for 20 min (EMD Millipore, Burlington, MA, USA).

Adipogenic differentiation began when cells reached 100% confluence. The medium was changed to MesenCult™ Adipogenic Differentiation Medium (Human, STEMCELL Technologies Inc.; Vancouver, BC, Canada); the day of this occurrence marked day 0 of differentiation. The differentiation medium was renewed every 3 days. Cells were fixed with 4% paraformaldehyde in PBS solution for 30 min and stained at 21 days of differentiation.

Chondrogenic differentiation was assessed in the form of a 3-dimensional (D) pellet culture system (5 × 10^5^ cells per pellet) as recommended by the manufacturer, using MesenCult™-ACF Chondrogenic Differentiation Medium (STEMCELL Technologies Inc.). On day 21, chondrogenic pellets were fixed with 0.1% glutaraldehyde for 20 min at RT, then washed twice with PBS. Pellets were embedded in Tissue-Tek® O.C.T. Compound (Sakura^®^ Finetek USA Inc., Torrance, CA, USA), cut into 7-μm sections, and stained with Alcian blue solution for 1 h at RT (Sigma Aldrich).

For keratocyte differentiation, CSSCs were plated at a cell density of 10^4^/cm^2^. When cells reached confluence, the medium was substituted with keratocyte differentiation medium that consisted of advanced DMEM (Gibco), antibiotics (penicillin, streptomycin, gentamicin), and L-ascorbic acid 2-phosphate (1 mM; Sigma Aldrich) and was supplemented with recombinant human TGF-b3 (1 ng/mL; R&D Systems) and recombinant human FGF2 (10 ng/mL; Abcam, Cambridge, UK). The differentiation medium was changed every 3 days.

Cell samples were collected for RNA analysis at days 0 and 7 of differentiation for chondrogenic and keratocyte differentiation and at day 21 for adipogenic and osteogenic inductions.

### 4.4. Immunofluorescence

Cells were fixed with 4% paraformaldehyde (Electron Microscopy Sciences, Hatfield, PA, USA) in PBS for 15 min at 4 °C and washed with PBS. The cells were then permeabilized with 0.2% Triton-X100 (Sigma Aldrich), and nonspecific antigens were blocked with 1% normal donkey serum (NDS) in PBS at RT for 30 min. Primary antibody incubation was performed overnight at 4 °C in a solution consisting of 0.2% Triton-X100 and 0.5% NDS. Antibody used was specific to adiponectin (mouse; 1:400; clone 19F1; Abcam). Donkey anti-mouse immunoglobulin conjugated to Alexa-Fluor 564 (1:400; Life Technologies) was used as the secondary antibody. Hoechst 33342 (ThermoFisher) was used for nuclear staining.

### 4.5. TNFAIP6 Expression

CSSCs were plated at a cell density of 10^4^/cm^2^. The next day, cells were treated with recombinant human tumor necrosis factor (TNF-α, 10 ng/mL; PeproTech, Rocky Hill, NJ, USA) and recombinant human interferon (INF-γ, 25 ng/mL; PeproTech) for 16 h. Samples were then collected for gene expression analysis of TSG-6 transcript.

### 4.6. Flow Cytometry

CSSCs were characterized by the expression of the MSC-specific surface antigens CD105, CD90, and CD73 and negative MSC markers CD 14, CD20, CD34, and CD45 using the phenotyping kit from Miltenyi Biotec (Bergisch Gladbach, Germany). The analysis was performed with a FACSAria flow cytometer (BD Bioscience, San Jose, CA, USA) at the flow cytometry core facilities at the Eye and Ear Institute of Pittsburgh and at the UCLA Broad Stem Cell Research Center. Fluorescence compensation settings were adjusted according to the kit’s instructions.

### 4.7. RNA Extraction, cDNA Transcription, and Real-Time PCR

RNA was isolated using the RNA Mini isolation kit (Qiagen Inc., Germantown, MD, USA). After RNA elution from the column, RNA was precipitated overnight with 450 mM ammonium acetate (Life Technologies), 200 ng/mL GlycoBlue™ coprecipitant (Life Technologies), and 100% ethanol (Sigma Aldrich). RNA concentration was measured by the NanoDrop™ 2000 spectrophotometer (v. 1.6.198, ThermoFisher), and 1 µg of RNA was used per transcription reaction using Superscript™ III Reverse Transcriptase (100 U/µg of RNA, Life Technologies). Primers for real-time PCR (RT-PCR) are listed in Appendix A. RT-PCR was performed with SYBR™ Green Master Mix (Applied Biosystems™, Carlsbad, CA, USA) in an Eppendorf Mastercycler^®^ ep realplex (v2.2, Eppendorf, Hauppauge, NY, USA). Two housekeeping genes were used per sample for normalization of the data. Crossing threshold (Ct) values were obtained. Absolute levels of transcripts were calculated with normalization of ΔCt, noted as 2^(−ΔCt)^.

### 4.8. Image Acquisition

Photos of cell morphology and immunohistochemical studies were taken using a fluorescence microscope (BZ-X710, Keyence Corporation, BZ-X Viewer version 01.03.00.05, Osaka, Japan).

### 4.9. Statistical Analysis

Statistical analysis was performed by using JMP^®^ Pro software (version 14.0.0, JMP Inc., SAS Institute, Lafayette, VA, USA). For two-pair comparison of the means, ANOVA test followed by Student’s *t*-test was performed. For multiple comparisons, ANOVA was followed by Tukey’s post hoc comparison of the means test. A *p*-value < 0.05 was considered statistically significant.

## Figures and Tables

**Figure 1 ijms-23-13399-f001:**
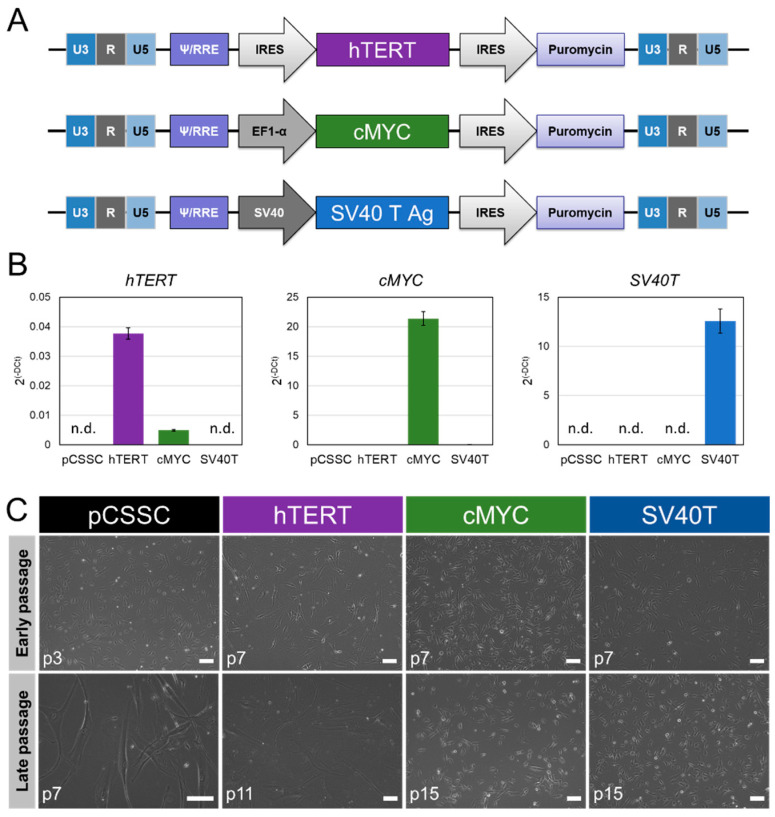
Generation of immortalized CSSC lines. (**A**) Schematic representation of lentiviral constructs for hTERT, c-MYC, and SV40T immortalizing methods. (**B**) Levels of transcript of the immortalization genes (hTERT (p7), c-MYC (p15), and SV40T (p15)) in each cell line, measured by RT-PCR. Results were expressed as mean ± SD. Statistical analysis: ANOVA comparison of the means, followed by all-pair comparisons Tukey’s test. CSSC: corneal stromal stem cells, pCSSC: primary CSSC; n.d.: not detected; p: passage number. (**C**) Phase-contrast images of pCSSC (p3 and p7), hTERT-imCSSC (p7 and p11), c-MYC-imCSSC (p7 and p15), and SV40T-imCSSC (p7 and p15) cells at early and late passages. Scale bar: 100 μm. CSSC: corneal stromal stem cell; pCSSC: primary CSSC; imCSSC: immortalized CSSC; p: passage number; SD: standard deviation.

**Figure 2 ijms-23-13399-f002:**
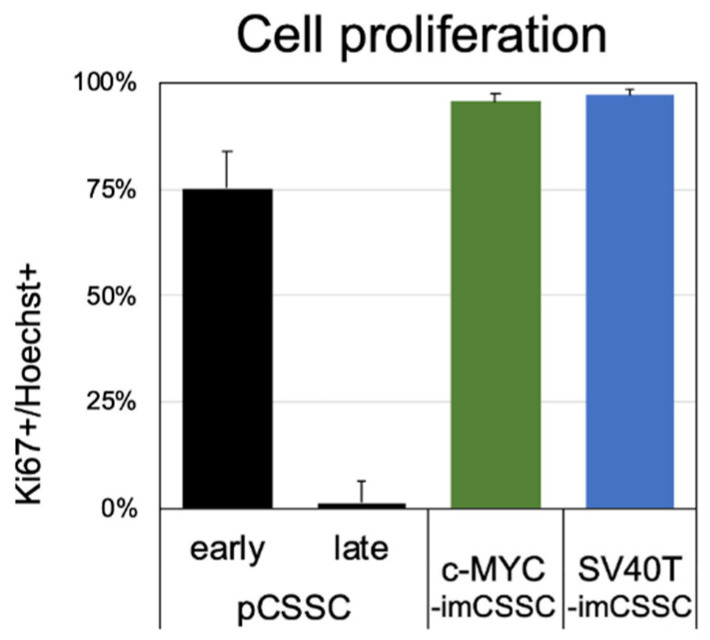
Cell proliferation of pCSSC and imCSSC cells. Immunostaining for the proliferation marker, Ki67, allowed quantification by computing the percentage of Ki67 positive cells against the number of nuclei (Hoechst staining). Passages 2–4 were used for pCSSC analysis at early passages; passage 7 was analyzed for pCSSC as late passages. Passages 10 to 14 were analyzed for the imCSSC. Results are expressed as mean ± SD. CSSC: corneal stromal stem cell; pCSSC: primary CSSC; imCSSC: immortalized CSSC; SD: standard deviation.

**Figure 3 ijms-23-13399-f003:**
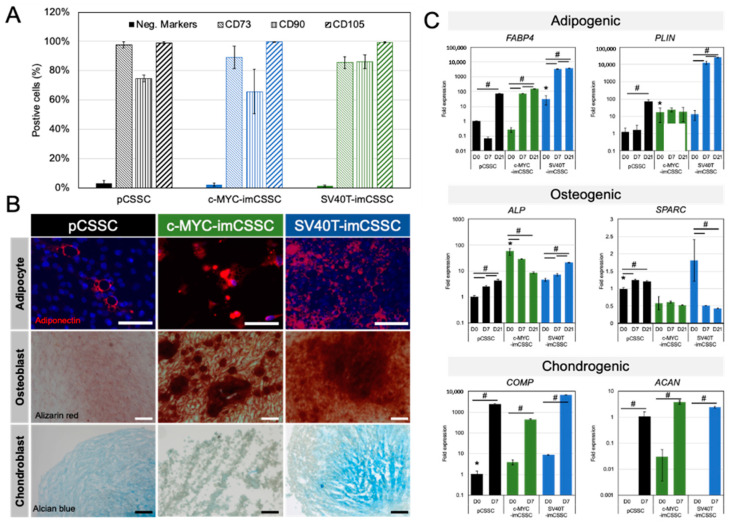
MSC phenotyping of pCSSC and imCSSC lines. (**A**) Analysis of MSCs’ surface antigens CD73, CD90, and CD105 and the negative markers (CD14, CD20, CD34, and CD45) was performed using flow cytometry (pCSSCs: p2 to 4; imCSSC: p7 to 12). Results are expressed as mean ± SEM. Statistical analysis: ANOVA test, not significant (*n* = 6 for pCSSC, *n* = 4 for c-MYC-imCSSC and SV40T-imCSSC). (**B**) Multilineage induction were visualized by staining in pCSSC (p3) and imCSSC (p10 to 12) after 21 days of differentiation. Adiponectin immunofluorescence staining was used for adipocyte lineage. Alizarin red staining allowed visualization of calcium depositions (osteoblast induction). Chondroblast lineage induction was shown by Alcian blue staining marking chondrocyte-associated extracellular proteoglycans. Scale bar = 100 μm (**C**) Multilineage differentiation of CSSC cell lines. Gene expression analysis of the adipose (*FABP4* and *PLIN*, top panel), osteogenic (*ALP* and *SPARC*, middle panel), and chondrogenic markers (*COMP* and *ACAN*, bottom panel) in each cell line tested by RT-PCR. Results are expressed as mean ± SD. Statistical analysis: ANOVA test, followed by all-pair comparisons Tukey’s. * *p*-value < 0.05 compared to pCSSC D0 (versus SV40T-imCSSC D0, c-MYC-imCSSC D0), # *p*-value < 0.05 compared to D0 (for each cell line). Scale bar: 100 μm. Neg. markers: CD14, CD20, CD34, and CD45; CD: cluster of differentiation; CSSC: corneal stromal stem cell; pCSSC: primary CSSC; imCSSC: immortalized CSSC; SEM: standard error of the means.

**Figure 4 ijms-23-13399-f004:**
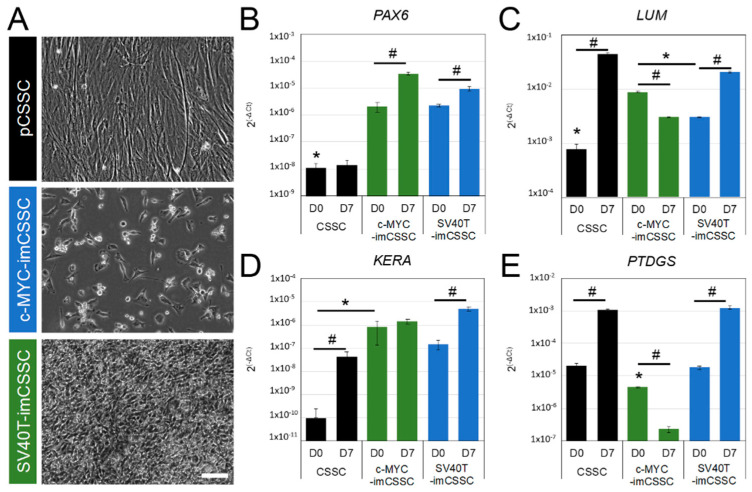
Differentiation capacity of generated CSSC lines towards keratocyte lineage. (**A**) Phase-contrast microscopy pictures of pCSSC and imCSSC lines after 7 days of keratocyte induction. Scale bar: 100 μm. (**B**–**E**) Expression of the keratocyte progenitor PAX6 (**B**) and mature keratocyte markers LUM (**C**), KERA (**D**), and PTDGS (**E**). Expression levels were assessed before and after 7 days of keratocyte differentiation by RT-PCR. Results were expressed as mean ± SEM. Statistical analysis: ANOVA followed by all-pair comparisons Tukey’s test, *: *p*-value < 0.05 between each cell line at D0; ANOVA followed by Student’s *t*-test comparison of means, #: *p*-value < 0.05 between control untreated (D0) and induced cells (D7) for each cell line. Passage 3 was used for pCSSCs and p12 to 14 for imCSSCs. CSSC: corneal stromal stem cell; pCSSC: primary CSSC; imCSSC: immortalized CSSC; SEM: standard error of the means; D: day.

**Figure 5 ijms-23-13399-f005:**
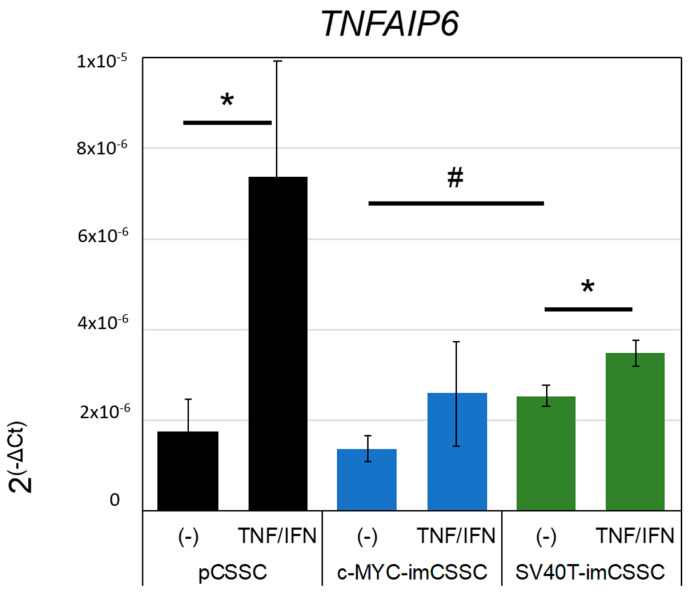
Anti-inflammatory properties. Anti-inflammatory response of pCSSC (p3 to 4), SV40T-imCSSC, and c-MYC-imCSSC cells (p8 to 12 for imCSSCs) after stimulation (TNF/IFN) with pro-inflammatory cytokines, IFN-γ (25 ng/mL) and TNF-α (10 ng/mL), as quantified by TSG-6 transcript levels (*TNFAIP6*) analysis as tested by RT-PCR. Results are expressed as mean ± SEM (*n* = 3). Statistical analysis: ANOVA followed by Student’s *t*-test comparison of means, * *p*-value < 0.05 between control untreated (−) and stimulated cells (TNF/IFN) for each cell line; ANOVA followed by all-pair comparisons Tukey’s test, # *p*-value < 0.05 between each cell line unstimulated. CSSC: corneal stromal stem cell; pCSSC: primary CSSC; imCSSC: immortalized CSSC; SEM: standard error of the means.

## Data Availability

Data can be made available from the corresponding author upon request.

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
