# Peer review of "Generation of Functional Immortalized Human Corneal Stromal Stem Cells"

_ijms, 2022, doi:10.3390/ijms232113399_

Round 1

Reviewer 1 Report

This study results in the development of immortalized cell lines of human cornal stromal cells. Therefore, it can be used in various studies on cornea stromal. However, some results show the possibility for the induced pluripotent stem cells. Therefore, the following additional data are considered to be needed.

-       Analysis for SOX2, OCT4, Nanog..

-       Data of cell growth and figure after long term culture

-       FACS analysis for CD45, CD34, CD11b, CK3

Author Response

Response to reviewer:
Dear Authors, It was a pleasure to read this manuscript. It is well written and the development of an immortalized human corneal stromal stem cells will be a great tool for drug discovery and mechanistic discovery, in the field of ophthalmology. The materials and methods are completed and it is possible to reproduce the experiments. After their isolation, human corneal stromal stem cells were immortalized using 3 different methods by introducing hTERT, c-Myc, and SVT40. Only the 2 last methods provided 2 stable cell lines. These 2 cells lines could proliferate over 10 passages, when the primary cells and the hTERT clone could not. SV40T and c-Myc imCSSC express specific mesenchymal stem cells markers and could differentiate into the 3 lineages, as reported by Dominici et al guidelines. In addition, SV40T and c-Myc imCSSC differentiated into keratinocytes. I have minor comments about the manuscript:

Author response: We thank the reviewer for the kind and positive comments regarding our manuscript. We have made our best effort to resolve the remaining points outlined below:

- The concentration of TNFa and IFNg used in the experiments should be added.
Author response: The concentration has been added into lines 107 and 231.

- Age of human tissue can affect their curative effect and their biological properties (e.g differentiation, proliferation, …), in addition of mutations on genes (pax6, keratins…). Do the authors know the age, health background of the donors or the authors could not choose the human corneal stromal stem cells?
Author response: We used the tissues available to us at the time of all experiments. We have added the following text into lines 55-56. “Human corneoscleral tissue from healthy donors aged 20 to 75 years old was used””

The authors should mention if the human corneal stromal stem cells used to engineer the 3 clones were from the same patient or from different patient?
Author response: They were from the same donor and this has been clarified on line 69.

Line 196: Did the authors mean Fig 1C, rather than Fig 3C?
Author response: We couldn’t identify this amendment but have carefully gone through the manuscript to make sure each figure is referred to correctly.

Line 374: ???? marks are at the end of a sentence. What is their meaning?
Author response: This has been changed in the manuscript on line 277. The “????” were referring to where the journal would publish the supplementary material.

- Reference in [X] should be placed before comas, before dots ending sentences. Please correct this.
Author Response: This has been corrected though out the whole manuscript.

Reviewer 2 Report

Dear authors.

The work presented here concerns the study of cells isolated from the corneal stroma. The focus of the study was the possibility of immortalizing cells isolated from the corneal stroma for use in other research work. 

As the authors note, there is no reproducible in vitro model for this type of research, which I can also confirm from my own experience. The paper is methodologically sound and the content represents a logical whole. In their research work, the authors confirmed the possibility of immortalizing a culture that retains the properties of the initial culture.

In my opinion, the presented work is an important contribution to science and the developed research model may help in the evaluation of the activity of novel drugs for use in ophthalmology in preclinical studies. 

In conclusion. I kindly request you create a separate chapter of conclusions, expanding it with detailed possibilities of using the obtained cells in scientific research and evaluating the activity of new drugs. 

Author Response

Response to reviewer 2

Dear authors.
The work presented here concerns the study of cells isolated from the corneal stroma. The focus of the study was the possibility of immortalizing cells isolated from the corneal stroma for use in other research work. As the authors note, there is no reproducible in vitro model for this type of research, which I can also confirm from my own experience. The paper is methodologically sound and the content represents a logical whole. In their research work, the authors confirmed the possibility of immortalizing a culture that retains the properties of the initial culture.
In my opinion, the presented work is an important contribution to science and the developed research model may help in the evaluation of the activity of novel drugs for use in ophthalmology in preclinical studies.
In conclusion. I kindly request you create a separate chapter of conclusions, expanding it with detailed possibilities of using the obtained cells in scientific research and evaluating the activity of new drugs.

Author response: We thank the reviewer for their positive review of our manuscript. We have snice made our best effort to resolve the remaining minor revisions as outlined below.

In conclusion. I kindly request you create a separate chapter of conclusions, expanding it with detailed possibilities of using the obtained cells in scientific research and evaluating the activity of new drugs.

Author response: The flowing has been added to the manuscript (Lines 272-778):
“The generation of these immortalized CSSCs will allow for significant advancement in the generation and translation of ophthalmic drug research. The capability to investigate drug-cell interaction with a reliable, reproducible cell line that functions equivalently to the primary cells from which it is derived is vitally important. Not only will these immortalized cell lines be beneficial in the field of drug discovery but they will also allow for further research in the regenerative and anti-scarring potential of CSSCs themselves. Without the need to continuously isolate primary cells, the regenerative potential of these immortalized CSSCs can also be investigated.”